“It’s okay because I’m just driving”: an exploration of self-reported mobile phone use among Mexican drivers

Useche Sergio A. sergio.useche@uv.es 1
Alonso Francisco 1
Faus Mireia 1
Cervantes Trejo Arturo 2
Castaneda Isaac 2
Oviedo-Trespalacios Oscar 3
1 Research Institute on Traffic and Road Safety (INTRAS), University of Valencia , Valencia , Spain
2 Faculty of Health Sciences, Anahuac University , Mexico D.F. , Mexico
3 Faculty of Technology, Policy, and Management, Delft University of Technology , Delft , The Netherlands
Abdullah Jafri
Electronic publication date: 2024 Feb 23
Publication date: 2024
Volume: 12
Electronic Location ID: e16899
Received 2023 Jul 20; Accepted 2024 Jan 16
Copyright: ©2024 Useche et al.
Copyright year: 2024
Copyright holder: Useche et al.
License: This is an open access article distributed under the terms of the Creative Commons Attribution License, which permits unrestricted use, distribution, reproduction and adaptation in any medium and for any purpose provided that it is properly attributed. For attribution, the original author(s), title, publication source (PeerJ) and either DOI or URL of the article must be cited.
License URL: https://creativecommons.org/licenses/by/4.0/

Keywords: Driving, Road distractions, Cell phone, Texting, Habits

Funding: AT&T Communications México Generalitat Valenciana ACIF/2020/035 The data collection process of this study was financially supported by AT&T Communications México as part of their social responsibility program. The development of this work was also supported by the research grant ACIF/2020/035 (MF) from “Generalitat Valenciana”. The funders had no role in study design, data collection and analysis, decision to publish, or preparation of the manuscript.

==============================
Introduction

Technological advancements have the potential to enhance people’s quality of life, but their misuse can have a detrimental impact on safety. A notable example is the escalating issue of distracted driving resulting from the use of mobile phones behind the wheel, leading to severe crashes and injuries. Despite these concerns, both drivers’ usage patterns and their risk-related associations remain scarcely documented in Mexico. Therefore, this descriptive study aimed to examine the mobile phone usage of Mexican drivers, its relationships to risk awareness and near-miss/crash involvement, and the self-reported underlying reasons for this behavior.

Methods

This cross-sectional study utilized a sample of 1,353 licensed Mexican drivers who took part in a nationwide series of interviews regarding their onboard phone use settings.

Results

A significant percentage of drivers (96.8%) recognize using a mobile phone while driving as high-risk behavior. However, only 7.4% reported completely avoiding its use while driving, with 22.4% identified as high-frequency users. Frequency was also found positively associated with the self-reported rate of near-misses and crashes. Furthermore, qualitative data analysis highlights the emergence of a ‘sense of urgency’ to attend to phone-related tasks in response to daily demands and life dynamics, offering a potential explanation for this behavior.

Conclusion

The results of this study suggest common patterns of onboard mobile use among Mexican drivers concerning driving situations and associated risks. This underscores the need for increased efforts to discourage onboard phone use in the country.

Introduction

Background

Mobile phones are beneficial devices that have transformed human communication and access to information. Indeed, many argue that mobile phones and similar technologies have altered how we live and socialize, leading to their constant use and integration into our daily routines (McEwan, 2020; Sivek, 2010). However, technology can also have undesirable and unintended consequences. For instance, the literature has highlighted matters such as the immediacy of response, novelty, unpredictability, and delocalization as latent technological side effects (Brooks, Longstreet & Califf, 2017; Silva, Suárez & Sierra, 2018). These adverse outcomes can be particularly concerning in safety-sensitive scenarios, with motor vehicle driving standing out among them.

Literature review

Mobile use while driving and its effect on their behaviors

The use of mobile phones is the most common cause of driver distraction, and research indicates that it can impair attention, perception, and executive functions beyond visual distraction, especially in high-stress or complex driving scenarios (Cosman et al., 2018; Valero-Mora et al., 2021; Ortega et al., 2021; Regan & Oviedo-Trespalacios, 2022). Recent systematic reviews have highlighted that almost all existing empirical literature suggests that using mobile phones while driving leads to crashes because it can divide the driver’s visual attention.

Recent studies (e.g., Stavrinos et al., 2018; Oviedo-Trespalacios et al., 2016) suggest that mobile phone-related tasks tend to divert drivers’ attention from the road ahead, resulting in the loss of important information. This reduction in driver attention increases the likelihood of crashes (Strayer, Watson & Drews, 2011). Moreover, recent research emphasizes that the distinction between primary tasks (such as physical vehicle operation) and secondary tasks (including mobile phone usage) is becoming less clear, leading to greater cognitive interference. This limits drivers’ attention-related resources and information processing capacity when performing simultaneous tasks (Wickens, 2008).

At a behavioral level, it is known that drivers who use their mobile devices behind the wheel are more prone to commit conventional traffic violations, including speeding (Hosking, Young & Regan, 2009) and stopping in inadequate places (Reimer et al., 2011). They also exhibit errors such as wrong detouring (Gaspar et al., 2014), poorer lateral control (Chen et al., 2022), spontaneous lane changes, and inadequate signaling (Kingery et al., 2015). The use of mobile devices while driving adversely affects drivers’ ability to make important decisions on the road (Collet, Guillot & Petit, 2010). Additionally, mobile phone use has been shown to increase drivers’ stress, anxiety, aggressiveness, and impulsivity (Chen, 2013; Berdoulat, Vavassori & Sastre, 2013).

Despite the common awareness of the aforementioned facts at a social level, dependence on connected devices continues to grow in modern society, which may generate a ‘sense of urgency’ to respond quickly to incoming requests, irrespective of their importance (Núñez & Zamora, 2017; Parasuraman et al., 2017; Prat et al., 2017). This heightened need to stay connected to one’s phone, even during tasks that demand full attention, such as driving, has been amplified (Bhattacharya et al., 2019; Oviedo-Trespalacios et al., 2018; Thomée, 2018; Oviedo-Trespalacios et al., 2019). Unfortunately, this elevated connection to mobile phones has often been associated with addiction and problematic use (Rahmillah et al., 2023). When examining crash data, distraction caused by mobile phone use emerges as a significant contributor to road accidents, more frequently than other sources of attention impairment, such as billboards, internal distractions, or various environmental stimuli (Ortega et al., 2021; Hinton, Watson & Oviedo-Trespalacios, 2022; Pöysti, Rajalin & Summala, 2005).

At the practical level, recent figures indicate an increase in the onboard use of mobile phones over the past few years, suggesting that information alone is not enough to alleviate the problem (Oviedo-Trespalacios, Truelove & King, 2020; Schwebel et al., 2012). Given this trend, there is a pressing need to implement policies and measures to discourage the use of mobile phones while driving and promote safe driving practices.

An examination of who uses and who does not use phones while driving

From a global perspective, onboard phone use is often linked to young, inexperienced drivers, increasing their crash likelihood by up to 400% compared to novice drivers not using phones while driving (Strayer et al., 2015). Similar trends are evident in other countries. In Australia, 36% of drivers over 25 reported phone use while driving, rising to 75% among those under 25 (Australian Transport Safety Bureau, 2005). In the USA, approximately 60% of surveyed individuals reported reading and/or writing text messages while driving, with a higher prevalence among young people (Gliklich, Guo & Bergmark, 2016). A national survey among American teenagers found that almost half of the participants had sent texts and/or emails while driving (Olsen, Shults & Eaton, 2013).

However, especially in developing countries, age-based trends in onboard phone use are becoming increasingly unclear and inconsistent. A growing number of adult drivers, particularly those in the working-age group, are using their phones while driving. Studies in countries like India and Vietnam indicate low levels of awareness and a rising trend in mobile phone use while driving (Chopdar et al., 2018; Useche et al., 2021; Nguyen-Phuoc et al., 2020). Empirical studies highlight shortcomings in enforcement efforts, suggesting that non-enforced regulations on mobile phone use while driving are insufficient (Ortega et al., 2021; Rudisill, Baus & Jarrett, 2019).

In Mexico, regulations on mobile phone use while driving exist (it is prohibited nationwide under the General Law of Mobility and Road Safety), and enforcement efforts have been increasing in recent years, albeit with limited success (United Mexican States, 2022; Oviedo-Trespalacios et al., 2017; Rudisill & Zhu, 2021). Indeed, a previous study conducted by Vera-López et al. (2013) showed that the use of mobile phones while driving is likely to increase in both cities and rural areas. However, no national research has been done to understand the current usage patterns and potential dangers. This information is unknown and uninvestigated. Therefore, this study can provide valuable insights and lessons for other Latin American countries facing road safety challenges, such as high crash rates, social inequalities, and a lack of road safety research, education, and training (Espinoza Molina et al., 2021; Faus et al., 2022; Faus et al., 2021; Haghani et al., 2022).

Study objectives and potential contributions

Bearing in mind the aforementioned considerations, this descriptive study aimed to examine the mobile phone usage of Mexican drivers, its relationships to risk awareness and near-miss/crash involvement, and the self-reported underlying reasons for this behavior:

1. Is driving while using a cell phone a frequent behavior among Mexican drivers?

H1: Some research suggests that that many drivers engage in this behavior because they perceive it as low-risk, mistakenly believing they possess the skill to use a cell phone and drive simultaneously without mishap (White, Eiser & Harris, 2004a).

2. Is there a relationship between frequency of use and the rate of accidents/near misses of drivers?

H2: It is expected a significant relationship between both variables, as numerous studies conducted in other countries have demonstrated a substantial correlation between the use of cell phones while driving and an increased crash risk. This is often attributed to the distraction of the driver’s attention caused by this task (Caird et al., 2014).

3. What are the key features used while driving?

H3: The motivations for using the phone while driving are diverse. Research conducted in the United States (Klauer et al., 2014; Tian & Robinson, 2017) and China (Zhou et al., 2012), among other countries, generally indicates usage for calls, messaging, GPS navigation, and social networks. Therefore, similar results are expected in Mexico.

4. In which road-related situations is phone use more prevalent?

H4: Certain situations seem to have a higher likelihood of triggering phone use-related behaviors. According to previous studies, these situations may include stopped traffic, such as at traffic lights or during traffic jams (Huth, Sanchez & Brusque, 2015; Ni et al., 2021).

5. What are the self-reported motives for using cell phones while driving?

H5: The reasons for using the telephone can be diverse, but scientific literature emphasizes risk perception as a moderating variable that increases or decreases the probability of engaging in this behavior. Individuals who perceive using the telephone while driving as a low-risk task are more likely to engage in the behavior, justifying it for reasons such as urgency or habit, among others (Cordellieri et al., 2022).

To address these questions, this study employed a mixed-methods design, incorporating both quantitative and qualitative analyses. To the best of our knowledge, no previous study conducted in the region, specifically in Mexico and Latin America, has addressed the issue using this methodological approach. In practical terms, this approach may contribute to strengthening policymaking and informing interventions aimed at reducing traffic crashes associated with the use of handheld devices while driving.

Materials & Methods

Design and procedure

The data reported in this study were collected through a National Survey aimed to encompass all age segments of the driving population rather than aiming for raw representativeness of the general population. The latter might differ in terms of age, occupation, driving patterns, and mobile phone usage (Nguyen-Phuoc et al., 2020; Kalantari et al., 2021). Therefore, this study employed a purposive sampling approach, a non-probability sample technique intended to collect data from individuals meeting specific features, known as inclusion criteria (Campbell et al., 2020). In this case, potential participants were personally approached at public locations and invited to participate in the study, provided they were licensed and active drivers with mobile phones, regardless of the brand and/or version. While this sampling procedure effectively represents a specific population of interest, it is important to note that the generalizability of the data must be carefully assumed to be limited to this particular group (Andrade, 2021).

Regarding the sample size, a minimum number of n = 680 participants was determined for a confidence level of 99% and a maximum margin of error of 1% (α = .010). However, due to a high participation rate (>70% of those contacted agreed to participate), the final sample consisted of 1,353 participants. Consequently, the number of participants is significantly higher than necessary to ensure representative results. The sample size calculation formula has been defined as follows: n=Zα2Npqe2N−1+Z2α2pq

Where:

N = population size

Z α = constant corresponding to Z α = 2.58 at 95% confidence level

e = sampling error.

p = proportion of individuals in the population that possess the characteristic under study (assumed by default as .50).

q = proportion of individuals who do not possess this characteristic (1-p; assumed by default as .50).

n = sample size

The survey was conducted through face-to-face personal interviews, utilizing a CAPI (computer-assisted interviewing) system on tablets. Interviews were recorded and geo-referenced to minimize both interview duration and potential data recording errors. Initially, potential participants were personally approached and invited to participate by a trained staff member. Upon agreement to participate, data collection was carried out using a semi-structured questionnaire.

Instrument development

The study questionnaire was developed based on a two-phase strategy, adhering to recommended practices commonly found in specialized literature (Boateng et al., 2018) and drawing from staff experience. The first phase involved instrument development, pinpointing key phone usage-related questions (e.g., frequency, reasons, etc.), their behavioral repertoires (i.e., mobile phone features feasible to be used while driving), and the road settings in which the phone is most commonly used while driving (e.g., slow/fast driving, red lights, etc.). These variables were selected based on existing scientific literature. Research conducted in the United States (Hill et al., 2015), Australia (Haque & Washington, 2015), or China (Zhang et al., 2019) highlighted traffic light stop situations and slow-speed situations such as traffic jams or intersections as the most frequent situations for cell phone use by drivers.

The question pool was examined and polished by all research staff members, and the language/terminology used was double-checked by an ad-hoc expert (with an extensive background in transportation studies) before conducting the preliminary piloting of the survey form.

In the second phase, the questionnaire was pilot-tested with a small sample of 18 volunteers. After responding to it, they were asked about any possible doubts or confusion regarding the questionnaire instructions, item contents, or vocabulary issues. In only a few cases, amendments were necessary (e.g., replacing the term “GPS navigation” with “map browsing”), and these changes were made to ensure a comprehensive understanding of the questionnaire contents and dynamics among a broad population-based set of respondent profiles.

Questionnaire contents

The research questionnaire was composed of two basic sections, described as follows:

The first section addressed basic demographic information, including age, sex, city/region, and basic driving habits. Additionally, it gathered information about the licensing status of participating drivers to ensure they were both active drivers and smartphone users.

The second part of the questionnaire aimed to address onboard phone usage patterns. Considering that all participants were known to be smartphone-using active drivers, the time criterion for usage patterns was set at the last 30 days (last month). Drivers’ mobile phone risk-related awareness was assessed through a Likert question, using a 1 (not at all) to 5 (high risk) scale. Similarly, the self-reported overall frequency of mobile use while driving was measured through a frequency-based item, using a scale with five response options ranging between 0 (never) and 4 (almost always/always I drive).

Onboard phone usage in common specific situations was assessed through a dichotomic 7-item scale covering four common driving settings: stopping at a traffic light, driving in intermittent traffic, driving at low speed, and driving faster than 25 miles or 40 kilometers per hour. Regarding internal consistency and reliability indicators, the scale showed a Cronbach’s alpha estimate of α = .824, a McDonald’s Omega of ω = .819, and a composite reliability index (CRI) of .941. All these values are above the commonly suggested .700 cut-off point in psychometric literature (Campbell et al., 2020).

Both primary tasks performed while onboard the phone and the core motivations for using phones while driving (in case of having reported doing so, at least with a low frequency) were open-questioned and subsequently grouped into categories. The most common primary tasks found were voice calling, reading texts (passive), composing texts or texting (active), map browsing (active use/programming of navigation apps), social networking, reading emails (passive), and writing emails (active). The cloud analysis of the self-reported motivations or reasons for using phones while driving was categorically analyzed.

Finally, the self-reported number of crashes suffered while using a mobile phone was assessed using the question: “Have you ever had a phone-enhanced driving crash?” with the following response options: No (43.9%), Nearly (i.e., a near miss; 14.8%), and Yes (16.4%). These three values were used to assess the relationship between current usage patterns and drivers’ likely phone-enhanced crash history.

Participants

This cross-sectional study used the data retrieved from a full sample of n = 1, 353 Mexican drivers with a mean age of 33.74 (SD = 11.6) years. The sample distribution was notably proportional to the driving population census distribution by age (adults; over 18 years old) and region of origin. Detailed age and sex-related data are presented in Table 1.

Table 1 Sex and age distribution of the study sample.

Variable	Value	Frequency	Percentage	
Sex	Woman	959	70.9%	
Man	391	28.9%	
Prefer not to say	3	0.2%	
Total	1,353	100%	
Age	Less than 25	406	30%	
26–30	284	21%	
31–35	145	10.7%	
36–40	90	6.7%	
41–45	115	8.5%	
46–50	128	9.5%	
Over 50	185	13.7%	
Total	1,353	100%	

Data processing

After meticulous data curation, open-question data were categorized, and all database variables were labeled. Descriptive analyses were then conducted to characterize Mexican drivers’ frequency and mobile phone use features. Bivariate associations among drivers’ age, mobile use frequency, and perceived risk were assessed using Pearson correlations. For qualitative analyses, data were labeled using a grouping categorical analysis strategy, following the steps suggested by Bergin (2018). This involved reducing data into emerging classes based on their contents. Textual data provided for semi-open questions were grouped by two qualified raters (SAU and OO-T), and codes were assigned to each categorical value. Since only one response had been provided by each participant, and the question was very concrete (i.e., their main motivation for onboard phone use), the overall inter-rater agreement was 92.4%, considered acceptable for this type of analysis, as the most common standard rounds the 80% agreement (O’Connor & Joffe, 2020).

Quantitative analyses were conducted using IBM SPSS (Statistical Package for Social Sciences), version 26.0, and Sigma Plot, version 12.0 for heat plots. The qualitative treatment of categorical (open question-based) variables was carried out using NVIVO software, version 11.0.

Ethics

To perform this study, the Ethics Committee of Research in Social Science in Health at the University of València was consulted, ensuring that it adhered to general ethical principles and conformed to the Declaration of Helsinki (IRB approval number: HE0002150421).

Concerning the data collection process, participation was anonymous and voluntary. Personal information was handled following applicable data protection laws and ethical guidelines. No financial rewards were offered to study participants. All participants provided written consent before taking part in the study, following an explanation of the research objectives and procedures by the research staff.

Results

Prevalence of the behavior in Mexican drivers

The descriptive results of this study help depict two essential facts in regard to mobile phone usage among Mexican drivers:

Firstly, self-reported awareness of the risk associated with using cell phones while driving shows high values. Approximately 23.6% of drivers consider this behavior “risky”, and 73.2% perceive it as a “high-risk” practice. Conversely, only 3.2% of the study participants view the risk of using a cell phone while driving as low. Using a 1 (not at all) to 5 (high risk) scale, the average value obtained for mobile phone risk awareness was M = 4.69 (SD = 0.54).

However (second fact), there is a significantly high frequency of in-vehicle cell phone use among Mexican drivers. Specifically, 22.2% of the sample reported always (6.4%) or almost always (15.8%) using the phone while driving. Additionally, almost half of the sample reported infrequent cell phone use (36.5% rarely and 34.1% sometimes). Only 7.4% of the drivers stated that they had never used a cell phone while driving in the last month.

Moreover, there were some significant Pearson bivariate correlations between (i) self-reported frequency of phone use while driving (in the last 30 days) and self-reported awareness of mobile phone risks (positive); (ii) drivers’ age and phone-risk awareness (positive); and (iii) drivers’ age and onboard phone use (negative). An extended set of descriptive data and bivariate correlations is presented in Table 2.

Table 2 Descriptive statistics and bivariate correlations among study variables.

Variable	Mean	SD 1	Range	Correlation	1	2	
1	Age	33.74	11.6	[18–78]	Pearson’s r	–	−.062*	
p-value	.023	
2	Onboard phone use (frequency)	2.77	1.01	[1–5]	Pearson’s r	−.062*	–	
p-value	.023	
3	Phone-risk awareness	4.69	.54	[1–5]	Pearson’s r	.078**	−.171**	
p-value	<.001	<.001	
Notes.

1SD Standard Deviation

* Correlation is significant at the .050 level (2-tailed).

** Correlation is significant at the .001 level (2-tailed).

Phone use frequency and crash involvement

Figure 1 presents a heat diagram comparing the density of phone use frequency crossed with the self-reported number of likely phone-enhanced/caused traffic crashes suffered by participants while driving. Each segment of the diagram represents a cross-section of the self-reported number of crashes (X-axis) with phone use frequency (Y-axis), with the magnitude (percentage) of coinciding cases depicted through heat patterns within segments. It is observed that the number of “not crashed” drivers is higher among those who report that they never or almost never use the phone while driving. Conversely, individuals who use the cell phone while driving with a certain frequency (sometimes, almost always, or always) are the ones who have been involved in traffic crashes the most.

Figure 1 Heat plot to assess the relationship between (self-reported) mobile use frequency and driving phone-enhanced crash record.

The heat plot shows a higher concentration of both near-misses and phone-enhanced crashes among drivers self-reporting a greater onboard phone usage. Bar/sector densities represent the magnitude of the associations presented in the figure, alternatively to the plot colors.

The crash history appears to be associated with the degree of onboard phone use, with 20% of drivers using it on a low-frequency basis self-reporting at least one phone-enhanced crash. This trend is similar in the case of frequent phone users behind the wheel, with 19% reporting involvement in at least one phone-caused crash. Conversely, this trend appears to be inverse among drivers not using phones while driving or using them very rarely, where phone-related crash involvement decreases to 12%. This difference is statistically significant with χ2 = 18.407; p < .001, potentially attributable to other causes.

Concerning self-reported near-misses (i.e., situations that almost ended in a crash, which could have been avoided at the last minute), it has been observed that frequent phone users report a higher rate of near-misses (27%) compared to both low-frequency users (25%) and, particularly, non-mobile phone user drivers (χ2 = 14.320; p < .001).

Commonly used phone features

Seen as a whole and regardless of their frequency, the most frequently self-reported phone features among Mexican drivers were map browsing (94.4%), reading income messages (88.8%), and texting (81.2%). On the other hand, the least prevalent mobile phone tasks included in the list were email writing (38.6%), voice calls (44.2%), and email reading (47.4%).

Secondly, considering the self-reported frequency of such risky mobile-related behaviors behind the wheel, Fig. 2 illustrates that, apart from being the most performed task, about 43% of map browsing users do it regularly (almost always/always). Furthermore, approximately one-quarter of drivers self-reported performing five of these tasks (i.e., reading messages, texting, social networking, reading/writing emails) on a very low-frequency basis. In contrast, only about two out of each 10 drivers read/write texts at a high frequency, and approximately only 11% regularly engage in social networking while driving.

Figure 2 Self-reported frequency of mobile phone feature use, organized from the most frequent (map browsing) to the least frequently reported (writing emails).

This plot shows how activities such as map browsing, reading messages (passive), and texting (active) remain the most commonly performed onboard mobile phone-related behaviors among Mexican drivers.

Driving situations involving the use of a cell phone

To evaluate the use of mobile phone features in common driving situations, drivers were asked to indicate whether they performed each of these tasks while: (i) stopped at traffic lights; (ii) driving in intermittent traffic; (iii) driving at low speed; and (iv) driving faster than <25 miles (40 kilometers) per hour.

Overall, the road setting most commonly referred as the main ‘enhancer’ of mobile phone use was, among the four, the typical situation of waiting at red traffic lights. In other words, one-third of both ‘message readers’ and ‘map browsers’ do so while stopping in front of semaphores. However, this percentage increases to almost 38% for voice calls, which appears to be the most widespread practice among drivers who use their mobile phones while behind the wheel.

Another noteworthy aspect is that voice calls are the only phone-related behavior consistently performed by more than 20% of mobile phone-using drivers while driving faster than 25 mph. Moreover, the frequencies of message reading are considerably higher than those obtained for texting-related tasks (i.e., both writing and responding to messages). The complete set of percentages concerning these four common road settings is available in Fig. 3.

Figure 3 Usage (yes/no) of different mobile phone features in four common road settings.

Note: Frequencies were retrieved only among drivers having previously stated to perform the behavior. Among the four driving situations or scenarios presented to participants, the most frequent for all the onboard phone-related tasks was while stopping at a traffic light. Accessibility note: The identification of the colors is not necessary to understand the content of the figure.

Attributed motives/reasons for mobile phone engagement

To hierarchically assess and understand the primary reasons Mexican drivers tend to use mobile phones while driving, the responses provided to this open question were categorized. The complete set of categories obtained from the study sample is visually represented in the word cloud presented in Fig. 4. Word cloud analyses aid in distinguishing the most frequently mentioned categories in participants’ responses, such as attributions, beliefs, or motives for engaging in certain behaviors. Their purpose is descriptive, and they lack inferential value (Rudisill, Baus & Jarrett, 2019).

Figure 4 Self-reported reasons for mobile phone use while driving among Mexican drivers.

Bigger words represent categories cited with a greater frequency. Among all the reasons attributed to using the mobile phone while driving, there stands out the perception of needing to attend it, a growing inclusion of mobile phone in daily-life dynamics, and a so-called “sense of urgency” raised among onboard phone-using drivers. The size of the words corresponds to their frequency in the participants’ speech (i.e., the larger the word size, the greater the number of times that it has been cited).

One of the key findings was that the only two terms exceeding 10% of participants’ quotations were “urgent” and “needed”, as in the cases of “it was because of a medical emergency for my son” (participant 254), “I needed to pick up the phone because it was an issue that could not wait” (participant 451), or “I needed to take the call; it was a work issue” (participant 625). This suggests a potential “sense of urgency” to use the mobile phone even while driving. Additionally, supporting the aforementioned, there was a considerable frequency of references to “everyday life” dynamics, the concern of not attending to the phone, and the “normalization” of this practice, as shown in words written in blue in the word cloud. Responses such as “because it is not a problem for me, I control the situation” (participant 98), “It’s okay because I’m just driving” (participant 236), “I always use my cell phone if I need it” (participant 965), or “it is not dangerous” (participant 1006) were expressed. On the other hand, there were some terms marginally quoted (<1% of participants), such as “myth”, “police”, and “not risky”.

Discussion

This study aimed to examine the mobile phone usage of Mexican drivers, its relationships to risk awareness and near-miss/crash involvement, and the self-reported underlying reasons for this behavior. Accordingly, its core findings and possible implications are discussed below:

The magnitude of the problem

At a descriptive level, our results suggest that many Mexican drivers continue to engage in mobile phone use while driving, despite acknowledging the associated risks. This finding aligns with studies in countries like Australia and the UK, suggesting that awareness alone may not lead to behavioral change (White, Eiser & Harris, 2004b; Truelove, Watson-Brown & Oviedo-Trespalacios, 2023). The pervasive integration of mobile phones into daily life could contribute to this phenomenon (Oviedo-Trespalacios et al., 2019).

However, it is worth noting that the self-reported methods in this study may underestimate the true extent of this behavior due to potential legal consequences and social desirability bias. Contrastingly, observational and proxy reports offer a more accurate depiction of mobile phone use while driving (Duarte & Mouro, 2019; Li et al., 2016; Useche, Faus & Alonso, 2022). This implies that the prevalence of mobile phone use while driving may be even more extensive than indicated in the present study. Moreover, mobile phone use stands as the most common driving distraction currently, with substantial changes not anticipated in the near future (WHO, 2011). This may pose challenges for researchers and policymakers, especially as for the common control and law-enforcement limitations documented by previous studies (Vera-López et al., 2013; Alghnam et al., 2019; Oviedo-Trespalacios et al., 2020).

Factors associated with mobile phone use while driving

We also investigated whether drivers who are aware of the risks associated with phone use while driving tend to avoid this behavior, aiming to identify potential intervention pathways. Despite a statistically significant negative correlation between self-reported awareness and the frequency of mobile phone usage, drivers do not rely solely on risk perceptions to abstain from this behavior. Previous studies suggest that, although drivers generally acknowledge the risks of using mobile phones while driving, systemic factors contribute to distraction, and drivers should not be solely held responsible for this behavior (Nees, 2019; González-Iglesias, Gómez-Fraguela & Sobral, 2015). Furthermore, the qualitative analysis findings support the assumption that individuals use their mobile phones to respond quickly and efficiently to contextual demands, highlighting the usefulness and integration of mobile phones in our lives.

Furthermore, phone use while driving was analyzed across four key road settings, revealing consistent patterns of behavior. Results indicated that most phone interactions occur while waiting at red traffic lights, with percentages ranging from 14% (reading emails, the least prevalent) to 38% (voice calling, the most frequently reported task). These findings align with previous observational studies, emphasizing that drivers actively seek low-complexity situations to reduce the risk associated with distraction (Amaya-Castellanos, 2020; Russo et al., 2018). Importantly, it remains unclear whether these self-regulatory behaviors have actual implications for safety. The study also found that, except for voice calls, drivers were more likely to use their mobile phones at low speeds than at high speeds, supporting results from other studies suggesting drivers reduce their speed as a compensatory strategy when engaging in distracting activities while driving (Ortega et al., 2021; Iio, Guo & Lord, 2021).

While the proportion is relatively small, texting while driving was somewhat prevalent in the Mexican case, ranging from one out of 100 drivers (1.1% at high speed) to one out of four drivers (26.5% at red lights). The increasing prevalence of texting while driving is a growing concern in recent years (Bergmark et al., 2016). This phenomenon may be attributed to a “sense of urgency”, where drivers are more likely to send or receive text messages while driving if they perceive the communication to be important, meaningful, or requiring a prompt response. These findings help highlighting the need for further exploration and discussion on how social dynamics may impact drivers’ technology-related decision-making and whether these behaviors contradict their available information (e.g., “it is forbidden”), or overcome their road risk awareness (Foreman et al., 2021; Useche, Alonso & Montoro, 2020).

Impact of mobile phone use while driving

The associations between onboard phone use and safety outcomes is also worth to discuss. Notably, there is a correlation between mobile phone use frequency and crash-related incidents, as well as a notable prevalence of near-misses associated with this behavior. However, it is relevant to acknowledge that the study’s data nature and item types preclude making causal inferences. A comprehensive approach, incorporating longitudinal methods and encompassing education and enforcement-based interventions, is likely imperative to effectively address this issue.

According to the study results, 73% of drivers reported either not using mobile phones while driving or using them very rarely, and no self-reported phone-related crashes occurred in this group. However, this percentage decreases to 55% and 54% for occasional and frequent mobile phone users while driving, respectively. The likelihood of drivers self-reporting mobile-enhanced crashes falls between occasional/mid-frequency (20%) and frequent (19%) users, but it hardly reaches 12% for drivers who use mobile phones very rarely. Remarkably, the relationship between the frequency of mobile phone use and crash likelihood may not be straightforward or linear, with previous research indicating potential significant increases in crash likelihood among phone-using drivers. Overall, the findings align with related studies, suggesting greater crash-related differences when comparing onboard phone users with those having limited exposure to mobile phone-related risky behaviors (Oviedo-Trespalacios et al., 2016; Oviedo-Trespalacios et al., 2018).

It is essential to note that near crashes, or near-misses, are distinct from actual crashes, often resulting from quick driver reactions or refocusing and serving as valuable indicators of potential crash risk or situational factors. The study findings reveal that near crashes effectively differentiate between mobile phone users and non-users, irrespective of usage frequency. Among drivers who rarely use their mobile phones while driving, only 15% reported experiencing a near crash. This percentage increases to 25% for occasional users and 27% for frequent mobile phone users while driving. These findings underline the relevance of near-miss incidents in understanding the safety implications of mobile phone use while driving.

Limitations of the study

Despite counting on an extensive and considerably balanced sample of participants, its cross-sectional design imposes limitations on establishing causality in the associations among study variables. Namely, it is important to highlight that:

Females had a higher degree of cooperation compared to males. While this does not impact the descriptive data provided by participants of both genders (minimum quotas were met), it suggests that male respondents’ hesitancy might also influence their responses to some extent (e.g., social desirability, acquiescence). Specialized studies have indicated that gender could have significant effects on issues addressed in this research, such as perceived vulnerability, norm awareness, and social behaviors (Djerf-Pierre & Wängnerud, 2016). Additionally, gender has been shown to account for notable differences in acquiescent and socially desirable responses concerning potentially sensitive topics for social groups (Djerf-Pierre & Wängnerud, 2016; Rammstedt, Danner & Bosnjak, 2017).

The exclusive reliance on self-report data raises concerns about the potential influence of common method biases, particularly considering the study’s focus on potentially sensitive behavioral issues (e.g., violations of laws or regulations), as noted in previous road safety studies (Useche, Alonso & Montoro, 2020). While questionnaire-based research is the norm for addressing such topics, it is advisable to interpret the outcomes with an awareness of the potential biases introduced by this method (Podsakoff et al., 2003).

The study’s limited number of presented road settings and the absence of task-related analytic procedures hinder the generalization of results. The inclusion of only four generic situations aimed to prevent survey length increase. Moreover, given the descriptive nature of the research and the study’s setting, the findings should be interpreted cautiously, recognizing their lack of inferential value.

Additionally, given the current high number and diverse nature of devices and diverse regional legislations, this study did not assess the role of hands-free devices. Future research could explore these specific issues, considering their known sensory interference with driving performance.

For future research, the reported findings can contribute to the design of more robust methodological approaches, such as longitudinal studies, experimental designs, or naturalistic observations using objective indicators of mobile phone usage and driving performance. These approaches would enhance our understanding of the emergence of technological distractions and their correlation with impaired driving performance.

Conclusions

The results of this descriptive study offer valuable insights into the prevalence and patterns of mobile phone use while driving in Mexico. The conclusions are formulated in response to the five key questions that guided the research.

• In relation to the first study hypothesis (frequency), the use of mobile phones while driving was found to be quite common. Only 7.4% of surveyed drivers self-reported never using them, while 70.6% were mid-intensity users, and 22.4% were very frequent (high-frequency) users.

• Secondly, a significant positive association was identified between the frequency of onboard phone usage and the self-reported rate of near-misses and crashes experienced by Mexican drivers.

• Thirdly, and as for specific phone features, map browsing, reading/writing texts, and social networking emerged as the most frequently performed phone-related features among drivers.

• Fourth, the most common road-related situations in which phone use was observed included stopping at traffic lights, driving in intermittent traffic, and driving at low speeds. However, a considerable number of drivers were still observed using their phones while driving at higher speeds.

• Finally, the study found that the most frequently cited reason for using a phone while driving was a perceived “sense of urgency”. Drivers reported feeling compelled to attend to mobile phone-related tasks in response to daily demands and the dynamics of life, shedding light on the motivations behind this behavior.

All in sum, the study’s results indicate common patterns of onboard mobile phone use among Mexican drivers concerning driving situations and risk-related associations. It is advisable for stakeholders, including policymakers and practitioners, to place increased emphasis on these findings. This can contribute to the development of effective measures and actions aimed at deterring drivers from using their phones while driving, ultimately helping to reduce distraction-related crashes in the country.

Supplemental Information

Supplemental Information 1 Draft questionnaire (English; Researcher form)

The English version of the root questionnaire used to conduct the research.

Supplemental Information 2 Draft questionnaire (Spanish; Researcher form)

The Spanish (i.e., originally applied) version of the root questionnaire used to conduct the research.

Data S1 Raw data

The authors wish to thank Arash Javadinejad, Ph.D. (licensed translator), for the professional edition of the final version of the manuscript.

Additional Information and Declarations

Competing Interests

Author Contributions

Human Ethics

Data Availability

The authors declare there are no competing interests.

Sergio A. Useche conceived and designed the experiments, performed the experiments, analyzed the data, prepared figures and/or tables, authored or reviewed drafts of the article, and approved the final draft.

Francisco Alonso conceived and designed the experiments, analyzed the data, authored or reviewed drafts of the article, and approved the final draft.

Mireia Faus analyzed the data, prepared figures and/or tables, and approved the final draft.

Arturo Cervantes Trejo conceived and designed the experiments, performed the experiments, authored or reviewed drafts of the article, data curation, and approved the final draft.

Isaac Castaneda conceived and designed the experiments, performed the experiments, authored or reviewed drafts of the article, field assessment, and approved the final draft.

Oscar Oviedo-Trespalacios analyzed the data, prepared figures and/or tables, and approved the final draft.

The following information was supplied relating to ethical approvals (i.e., approving body and any reference numbers):

Ethics Committee of Research in Social Science in Health of the University of Valencia (IRB approval number: HE0002150421).

The following information was supplied regarding data availability:

The raw data is available in the Supplemental File.

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
