# Peer review of "“It’s okay because I’m just driving”: an exploration of self-reported mobile phone use among Mexican drivers"

_PeerJ, doi:10.7717/peerj.16899_

## Round 0.1 · original submission · Major Revisions

Please pay special attention to the methods section and make necessary revisions to the manuscript in accordance with the feedback provided by the reviewers.

**Language Note:** The review process has identified that the English language must be improved. PeerJ can provide language editing services - please contact us at copyediting@peerj.com for pricing (be sure to provide your manuscript number and title). Alternatively, you should make your own arrangements to improve the language quality and provide details in your response letter. – PeerJ Staff

Reviewer 1 ·

Basic reporting

1. Numerous grammatical and typological errors are present in the manuscript. Authors are strongly urged to diligently rectify these errors during the revision process.

2. While the introduction is comprehensive, it lacks coherence. Although similar research has been conducted globally and in Mexico, the authors fail to establish the significance and novelty of their current study.

3. The stated objectives of the study pertain to investigating drivers' mobile phone use, its correlation with risk awareness, crash involvement, and understanding underlying reasons. However, the authors have not adequately delved into prior literature in these domains, instead mostly focusing on risks associated with phone use while driving. A restructured introduction is recommended, solidifying the study's objectives and rationale.

4. Figures are straightforward and effective, yet their presentation in the results section could be improved by either simplifying complex terms or providing explanations within the text.

Experimental design

1. As indicated, this original primary research employed field data collection from Mexican drivers. However, the manuscript lacks mention of the psychometric properties of the survey questionnaire used. While a pretest was conducted, the final questionnaire's psychometric properties should be included.

2.Clear clarification is needed regarding the determination of sample size, including the population base and methodology employed. The increase in sample size from n=680 to n=1353 requires detailed explanation.

3.The section detailing the questionnaire's content is overly lengthy. As the questionnaire itself is provided with the manuscript, it is advisable to include only essential details here, avoiding the inclusion of the entire questionnaire content.

4. Given the study's objectives, the rationale for not utilizing more sophisticated analysis techniques should be addressed. The chosen methods lack sufficient explanation and justification for their selection.

Validity of the findings

1. The rationale behind survey design choices and questionnaire items requires clearer elucidation. For instance, the options selected for "road settings most commonly comprising it in literature" lack proper referencing and seem arbitrarily chosen.

2. The results presentation is lacking coherence. It is recommended that the authors thoroughly revise this section, ensuring a strong connection with the study's objectives and eliminating redundant data points already presented in tables/figures.

3. The Chi-square results table requires enhancement. Clear labeling of variables within the table is recommended.

4. The authors fail to support the novelty and relevance of the study and have kept the study very minimal. The entire study explored very limited aspects of the objective mentioned in the study.

Additional comments

1. The study could benefit from more in-depth exploration of the issue. Which means including more questions on the mentioned objectives in the survey design.

2. As mentioned in the methods part, the study is descriptive in nature and has two open ended questions, the results of which are presented in the form of word cloud. However, it would really benefit the claims if authors could present some quotes from the respondents while explaining the results in discussion section. Much like how they have used a quote in the title of the study.

Reviewer 2 ·

Basic reporting

1. The initial part lacks proper structure and connection. The content seems scattered, with repeated information spread throughout different parts, making the logical progression unclear. Also, certain parts seem like a mix of loosely related sentences, leading to confusion and lacking a central theme. Though the author's intent is apparent, the way it's presented hampers the overall effectiveness of the introduction. To improve this section, a thorough review is suggested. The focus should be on reorganizing the content more orderly, allowing the main ideas to be communicated clearly and smoothly.

2. In the title you've provided, "It's okay because I'm just driving": An exploration of self-reported mobile phone use among Mexican drivers, "It's okay because I'm just driving" seems to imply a direct quotation from the participants, but the data you provided is not competent to it. However, if the quoted statement is not directly supported or backed up by the data used in the study and is merely illustrative, it could be misleading to include it in the title. Titles are expected to represent the content and findings of the manuscript accurately.

3. Many jargon have been used in the entire manuscript. It is understandable that the manuscript has been translated from another language to English; hence, certain words have been lost in translation. Authors are requested to carefully revise the manuscript and minimize the use of jargon and wrong terminologies.

4. The organization of various elements of the manuscript is flawed. The authors are advised to limit the word count and keep the manuscript as concise and cohesive as possible.

5. Table 2 which represents the results of the Chi-square test is not well made. The authors have failed to mention the particulars of the table properly. Please reconstruct the table again and mention each particular clearly.

Experimental design

1. The authors indicate that similar studies have been conducted in other low- and middle-income countries (LMICs) and within the same country. However, they have not effectively conveyed the novelty and originality of their study, how it differs from other works, and what it adds to the existing literature.

2. How was the sample size selected? Please elaborate on it and the reason and method for increasing the sample size to almost double of the previously decided sample size.

3. The authors are requested to provide the psychometric properties of the survey tool utilized. They have mentioned doing a pilot test of the survey, although have not talked about its psychometric properties in the manuscript.

4. From lines 274-276, the authors have mentioned that “as expected, self-reported mobile phone related risk awareness in driving presents high values” followed by a contradictory sentence “with 3.2% of the study participants considering that the risk implied by phone use while driving is low”. This sentence frame is contradictory and ill placed. In the following sentence it is reported that 73% reported it as high risk. Why not use this line within the previous sentence instead what originally has been used. This is an example of lousy writing and reporting throughout the results section. Please revise the results section thoroughly.

5. One of the commonly used phone features has been taken as “Navigation/Map Browsing”. How is it taken as a proxy for harmful/high risk phone use when it might be an essential use of phone during driving. It was also found to be the most sought reason for mobile use by drivers. It is essential for many drivers to use such technology for safe and efficient travel. Please elaborate on how it results in potential risk during driving.

6. From lines 211-214, the specific situations during which mobile phones could be used are defined. Authors have not explained the reason to choose these parameters among so many others. Are there any previous studies which have used same/similar parameters or is there any traffic rules in Mexico based on which these parameters were chosen. Please elaborate on this.

Validity of the findings

1. The provided questionnaire and analytical process is very surficial i.e. does not delve into the aimed objectives of the paper.

2. The results and discussion sections of the manuscript is also unnecessarily lengthy. Authors are requested to keep the discussion on point and use appropriate references.

---

## Round 0.2 · Minor Revisions

I appreciate the efforts made to enhance the content of your manuscript. However, there is a need for further improvement in the writing style, with a focus on conciseness and clarity as suggested by the reviewers. Please provide an English translation for the sentence in Spanish and consider incorporating a formula for sample determination in the methodology section. While the conclusions align well with the findings, attention is required to rectify grammatical errors and ensure adherence to APA citation style. A comprehensive revision is recommended to enhance academic coherence.

Reviewer 1 ·

Basic reporting

1. The writing style of the paper can be improved and at places can be made less wordy. Overall authors have improved upon the content of the manuscript. However better reporting is expected with new version. I would recommend authors to keep the manuscript brief, to the point and less wordy.
2. Under heading Study Objectives and Potential Contribution, under point 4 the sentence ends with "these could be situaciones de tráfico detenido como semáforos o atascos (54,55)." Please provide a translation of the sentence in English for clarification.

Experimental design

1. The authors have improved upon the reporting of their methodology and their tools.

Validity of the findings

1. With review, conclusions are well stated and have been tied to the findings of the study.

Additional comments

1. Overall, writing of the manuscript can be improved. There are a few noticeable grammatical errors. On the same note, I invite authors to recheck their references as I could notice a few mistakes such as APA style citations in a few places. Please rectify these errors.

Reviewer 2 ·

Basic reporting

Overall, the manuscript seems improved. But in terms of academic cohesive writing, comprehensive revision is required.

Experimental design

In line no. 179-180, it is mentioned that "the minimum number of participants needed to be representative was calculated." It is suggested to mention the sample determination formula according to the study design.

Validity of the findings

no comment

Additional comments

no comment

---

## Round 0.3 · accepted · Accept

Thank you for your revised manuscript which has been accepted.